# Promotion Effect of Palladium on BiVO₄ Sensing Material for Epinephrine Detection

Hsiang-Ning Luk [1] , Tsong-Yung Chou [2], Bai-Hao Huang [3], Yu-Syuan Lin [3], Hui Li [4] and Ren-Jang Wu [3,*]

[1] Department of Anesthesia, Hualien Tzu-Chi Hospital, Hualien 97002, Taiwan; lukairforce@gmail.com
[2] Department of Laboratory Medicine and Biotechnology, Graduate Institute of Medical Biotechnology, Tzu Chi University, 701, Zhongyang Road Section 3, Hualien 97004, Taiwan; cty@mail.tcu.edu.tw
[3] Department of Applied Chemistry, Providence University, Shalu, Taichung 43301, Taiwan; s1060529@gm.pu.edu.tw (B.-H.H.); g1090031@gm.pu.edu.tw (Y.-S.L.)
[4] Center for Advanced Thin Films and Devices, School of Materials and Energy, Southwest University, Chongqing 400715, China; lihui@kaist.ac.kr
[*] Correspondence: rjwu@pu.edu.tw; Tel.: +886-4-26328001-15212; Fax: +886-4-2632-7554

**Abstract:** In this study, the Pd/BiVO₄ composite was prepared by hydrothermal method as an electrochemical sensing material for epinephrine. X-ray diffraction, scanning electron microscopy, and a transmission electron microscope were used to characterize the samples. In the electrochemical detection system, cyclic voltammetry and differential pulse voltammetry were applied to measure the concentration of the epinephrine solution (0.9–27.5 μM) with the Pd/BiVO₄-coated glassy carbon electrode. As a result, the oxidation peak current of Pd/BiVO₄/GCE demonstrated good linearity with the epinephrine concentration. The detection limit of the epinephrine concentration by cyclic voltammetry and differential pulse voltammetry were 0.262 μM and 0.154 μM, respectively. Additionally, the proposed sensing material exhibited good reproducibility, stability, and selectivity. A plausible sensing mechanism was proposed.

**Keywords:** epinephrine; cyclic voltammetry; differential pulse voltammetry; Pd/BiVO₄

## 1. Introduction

Epinephrine (EP) is the neurotransmitter material essential for human body functions. The chemical structure of EP (adrenaline) is illustrated in Figure 1a. Generally, EP is one of the most important neurotransmitters, which plays a vital role in the transmission of nerve impulses. Numerous diseases are related to abnormal EP concentrations, such as Parkinson's disease, Alzheimer's disease, and stress and thyroid hormone diseases. Moreover, EP is used as an emergency medicine to treat seizures with cardio tissues. Thus, the accurate quantitative determination of EP is necessary for diagnoses of related diseases and the preparation of intravenous solution for clinical practice [1,2]. There are various ways to determine the EP concentrations, such as high-performance liquid chromatography-mass spectrometer (HPLC-MS) [3], fluorescence spectrometry [4], capillary electrophoresis [5], and electrochemical detection [6,7]. Among them, the electrochemical system exhibits advantages of low cost, easy fabrication, rapid detection, and easy operation.

The electrochemical sensors have good properties due to their high surface area and good electrical conductivity [6–9]. Especially in the past decade, there have been many nanomaterials applied in electrochemical measurement for EP detection, such as CuFe₂O₄, Au, graphene, polyaniline, Au-Pd decorated reduced graphene oxide, and so on [8–12]. Table 1 summarizes the sensing properties of some reported electrochemical sensors [8–11]. A nanostructured Au electrode was applied for sensing a wide range of epinephrine concentrations by linear sweep voltammetry (LSV) and differential pulse voltammetry (DPV) methods [8]. The anodic aluminum oxide structure depositing with a highly ordered Au film was served as a sensitive electrode for a simple and fast electrochemical determination

of epinephrine. The LSV and DPV have shown good linearity with the epinephrine concentration range of 60–600 µM, and 10–150 µM, respectively. The tetrahexahedral Au-Pd core–shell on the reduced graphene oxide was prepared for epinephrine detection [9]. The cyclic voltammetry (CV) and DPV methods were applied, and consequently presented a lower limit of detection to 0.0012 µM, and wide linear detection ranging from 0.001 µM to 1000 µM. A modified electrode based on graphene quantum was prepared for application to the electrochemical detection of epinephrine; the detection concentration range was measured from 0.36 to 380 µM [10]. The gold nanoparticle-polyaniline nanocomposite was prepared onto the surface of the glassy carbon electrode (GCE) to fabricate a voltammetric sensor for epinephrine detection [11]. The detection concentration range was measured from 0.4 to 10 µM, and the detection limit was obtained as 0.08 µM.

**(a)**

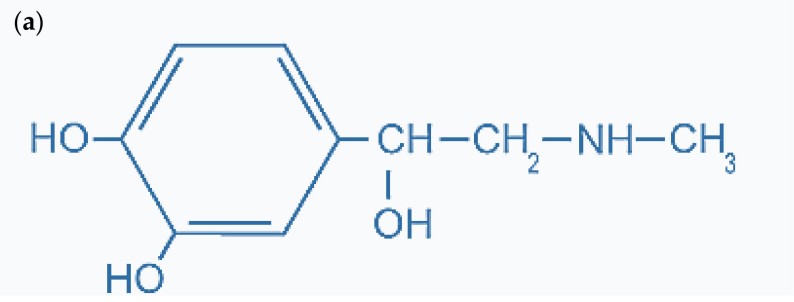

**(b)**

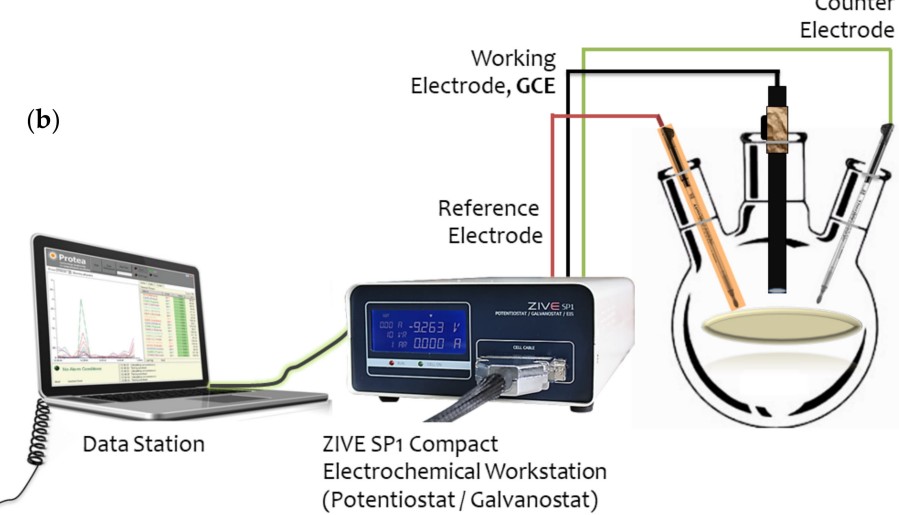

**Figure 1.** (**a**) Chemical structure of epinephrine, and (**b**) Electrochemical detection system for epinephrine.

**Table 1.** Comparison with epinephrine sensing properties on various sensing materials.

| Author/Year/Reference | Sensing Material | Range of Detection (µM) | Detection Limit (µM) |
|---|---|---|---|
| E. Wierzbicka /2016/[8] | Nano Au/high ordered anodic $Al_2O_3$ | 10–150 | 3 |
| W. Dong/2018/[9] | Au-Pd@reduced graphene oxide | 0.001–1000 | 0.0012 |
| J. Tashkhourian /2018/[10] | Graphene | 0.36–380 | 0.16 |
| L. Zou/2018/[11] | Au-polyaniline | 0.4–10 | 0.08 |
| This work | 1% Pd/BiVO$_4$ | 0.9–27.5 | CV: 0.262 ($R^2$ = 0.9998) DPV: 0.154 ($R^2$ = 0.9998) |

$ABO_4$-type metal oxide exhibits efficient electrocatalytic behavior toward EP. Therefore, $ABO_4$ has great research and development potential as an electrochemical sensor with good sensitivity [13]. Recently, $BiVO_4$ has been applied as a supercapacitor material in energy storage systems, photoanode material in water pollution monitoring, and photocatalyst in wastewater treatment owing to its narrow bandgap (2.4 eV), non-toxicity, good chemical stability, and outstanding photocatalytic response with low production cost [14,15]. Moreover, it can be easily doped with other material to improve the electrocatalytic properties [16]. $BiVO_4$ with modification shows a good future in electrochemical applications. However, the application of $BiVO_4$ for electrochemical detection of EP has been barely reported.

Therefore, the object of this work is to develop a simple prepared $BiVO_4$-based electrochemical sensor for EP detection. The electrochemical detection system is displayed in Figure 1b. The fabricated Pd doped $BiVO_4$ was prepared onto a carbon paste electrode as the working electrode. The material characterization and the electrochemical behavior of the $Pd/BiVO_4$ composite were systematically studied. Moreover, the possible EP detection mechanism was discussed. The proposed $Pd/BiVO_4$ is a promising sensing material for EP detection with a wide range and low detection limit as presented in Table 1.

## 2. Results and Discussion

### 2.1. Characterization of the Sensing Materials

Figure 2 presents the XRD patterns of (a) $BiVO_4$, (b) 0.5% $Pd/BiVO_4$, (c) 1% $Pd/BiVO_4$, and (d) 2% $Pd/BiVO_4$, respectively. The main diffraction peaks at 2θ of 18.7°, 28.6°, and 30.5° are assigned to the (110), (121) and (040) crystal planes of monoclinic scheelite $BiVO_4$, respectively [15]. All the diffraction peaks correspond to the monoclinic scheelite $BiVO_4$ phase (JCPDS No. 14-0133) regardless of the Pd loading. Neither the peak attributed to the Pd species nor the significant peak shift were observed in Figure 2, indicating no effect of the presence of Pd on the crystalline lattice of the $BiVO_4$ due to the low amount of the loaded Pd.

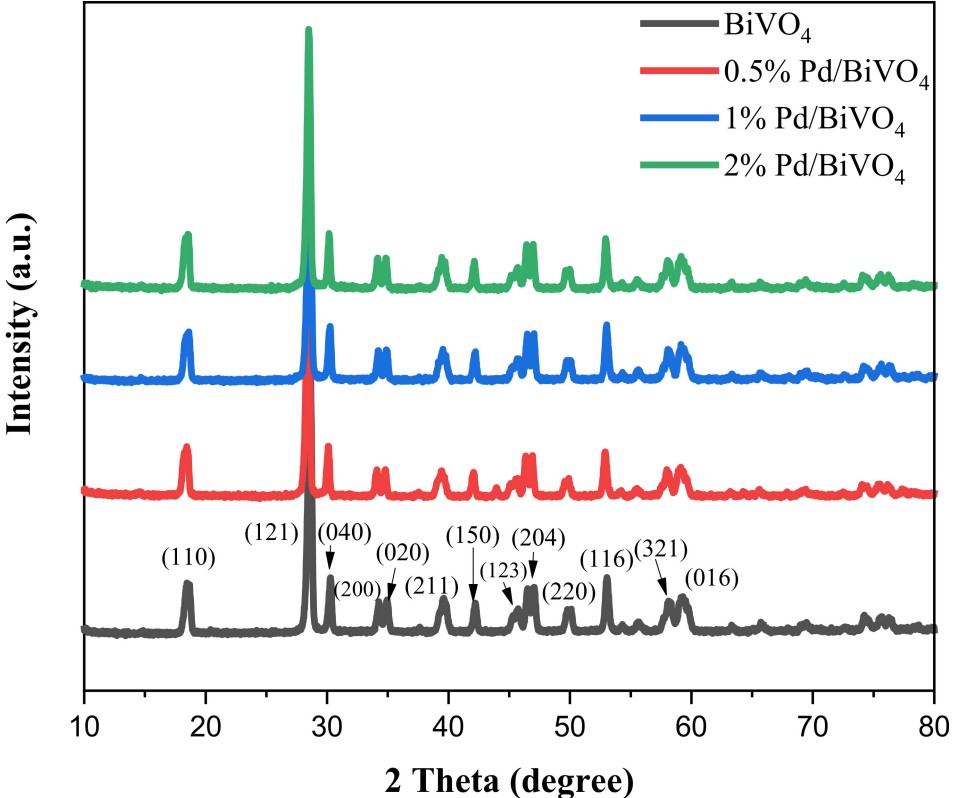

**Figure 2.** XRD patterns of obtained samples.

Figure 3 shows SEM images of (a) $BiVO_4$, (b) 1% $Pd/BiVO_4$ with magnification ×5000, and (c) of 1% $Pd/BiVO_4$ with magnification ×20,000. The SEM images present well crystallized samples possessing a uniform morphology of rod and sphere-like structures with the agglomeration of particles. The grain size was around 100 nm. After Pd doping, there appears to be spots attached to the surface of the $BiVO_4$ plates in Figure 3b. Furthermore, this phenomenon is clearly observed in Figure 3c with larger magnification (×20,000), revealing the formation of Pd nanoparticles. This result indicates the successful loading of Pd nanoparticles to afford $Pd/BiVO_4$ with relatively high dispersion by the low-cost and simple hydrothermal method.

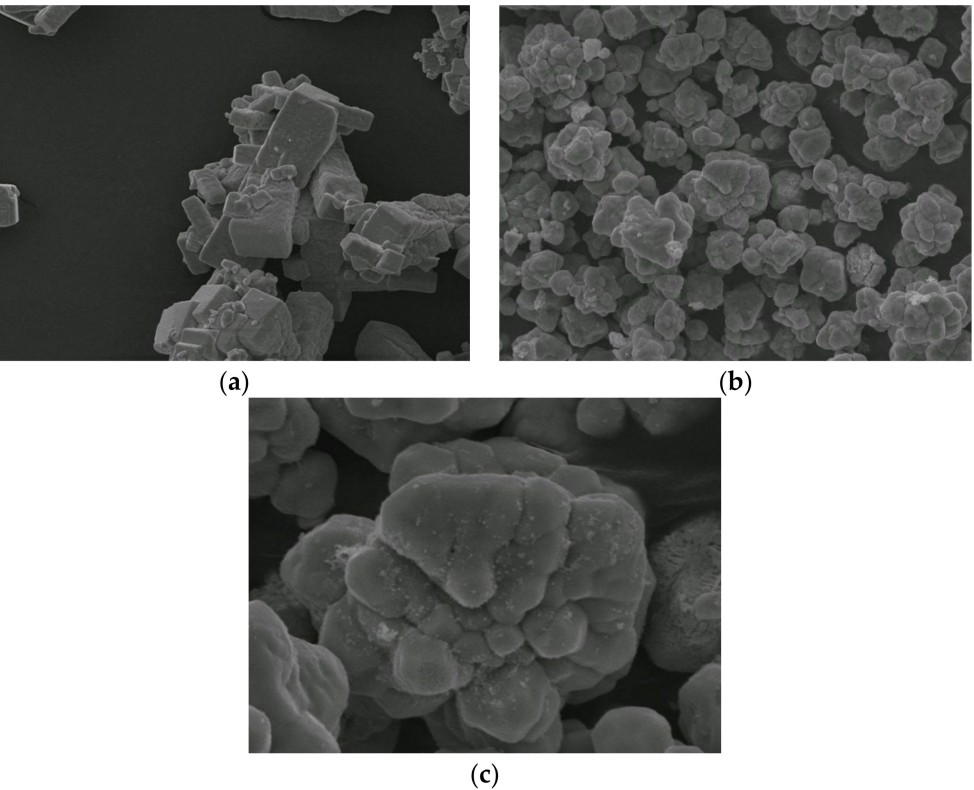

(a)  (b)

(c)

**Figure 3.** SEM images of (**a**) $BiVO_4$, (**b**) 1% $Pd/BiVO_4$ (×5000), and (**c**) 1% $Pd/BiVO_4$ (×20,000).

TEM images of 1% $Pd/BiPO_4$ are displayed in Figure 4a,b, and the Pd nanoparticles loaded on the surface of $BiVO_4$ are observed. It seems that $BiVO_4$ has palladium not only on the surface but also in the inner structure. Figure 4a shows a nearly uniform distribution of black spots. In the TEM image with a larger magnification (Figure 4b), the crystalline lattice space obtained from the line width was 0.29 nm, which was identified as the (040) crystal plane of $BiVO_4$. In addition, a 0.23 nm interfringe distance was observed, which was close to the lattice spacing of the (111) crystal plane of Pd. This result indicates the high dispersion of the Pd nanoparticles which could favor the electron collections. The elemental composition of the synthesized 1% $Pd/BiVO_4$ was elucidated using energy-dispersive X-ray spectroscopy analysis, and the result is shown in Figure 4c. Pd, Bi, V, and O elements were detected from the proposed $Pd/BiVO_4$ composite. This EDS result reveals that the obtained $Pd/BiVO_4$ does not contain other impurities. Combined with the aforementioned TEM and XRD results, the successful fabrication of high-purity $Pd/BiVO_4$ composite material can be confirmed.

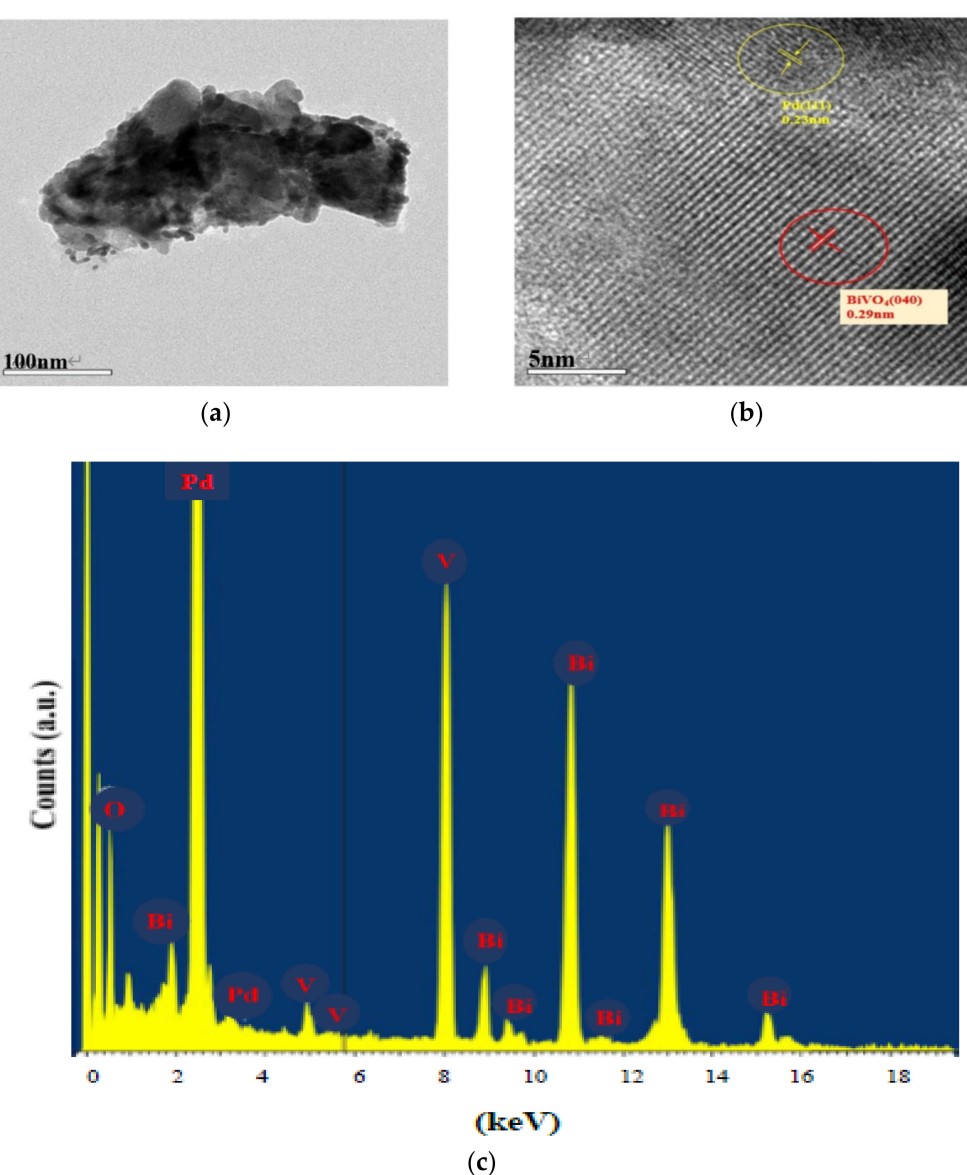

**Figure 4.** (**a**) TEM images of 1%Pd/BiVO$_4$; (**b**) TEM images of 1%Pd/BiVO$_4$ with high magnification, and (**c**) EDS spectrum of 1%Pd/BiVO$_4$.

### 2.2. Electrochemical Sensor Behavior

A cyclic voltammetry (CV) quantitative test was conducted using phosphate buffered saline solution (PBS) as the solvent, and a bare glassy carbon electrode, BiVO$_4$ and 1% Pd/BiVO$_4$ coated glassy carbon electrode as the work electrode, respectively. The potential range was set from −0.8 V to 0.8 V with a scan rate of 1.0 mV·s$^{-1}$. The test results are displayed in Figure 5A; 1% Pd/BiVO$_4$ exhibited better electrochemical performance with a significantly higher redox peak current than those of bare GCE and BiVO$_4$. The electrochemical activity of different modified electrodes with various Pd loading amounts were examined by CV in 27.5 μM epinephrine (Figure 5B). The obtained 1% Pd/BiVO$_4$ processes the highest redox peak current value as I$_{pa}$ = 6.0 μA and I$_{pc}$ = −12.0 μA, which means a better electrochemical performance for EP detection than 0.5% Pd and 2% Pd on BiVO$_4$. Appropriate amounts of addition Pd can promote the sensing signals; in this study, Pd existed not only on top of the structure but also sometimes in the inner structure. The overloading palladium of 2% might block the surface-active sites of BiVO$_4$. It should be noted that the potential of 0.40 V, where the oxidation peak current appeared, could be identified as the potential of oxidation action of epinephrine to the quinone [17].

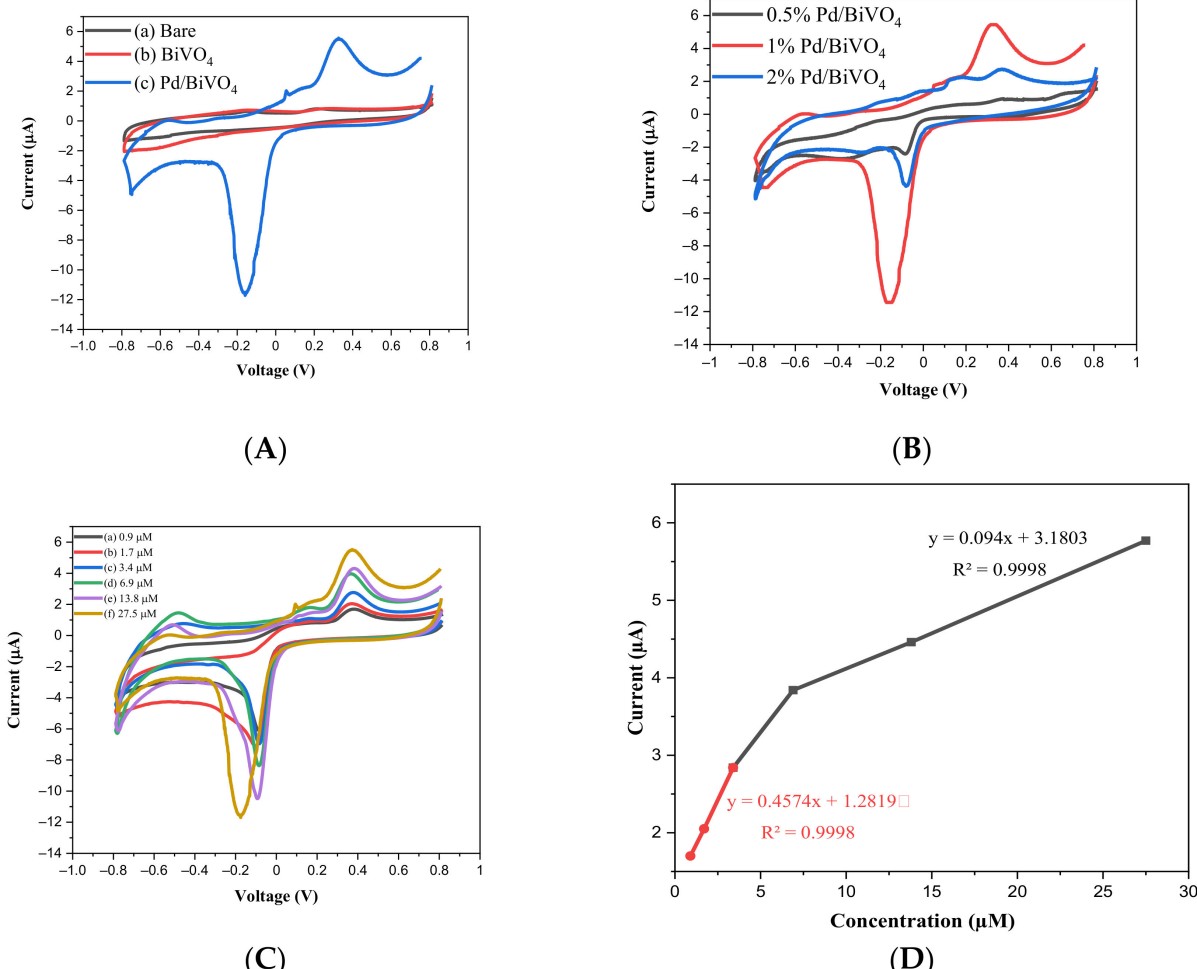

**Figure 5.** (**A**) CV response of (a) bare GCE (b) $BiPO_4$/GCE (c) 1%Pd/$BiVO_4$/GCE at scan rate of 1.0 mVs$^{-1}$; (**B**) CV response of 27.5 μM epinephrine at Pd/$BiVO_4$/GCE with various Pd amounts; (**C**) CV response at 1%Pd/$BiVO_4$/GCE with various epinephrine concentrations (a) 0.9 μM (b) 1.7 μM (c) 3.4 μM (d) 6.9 μM (e) 13.8 μM (f) 27.5 μM; (**D**) The plot of $I_{pa}$ against various epinephrine concentrations at 1%Pd/$BiVO_4$/GCE.

The effect of the variation of epinephrine concentrations on the sensing performance of 1% Pd/$BiVO_4$ was studied. PBS with epinephrine concentrations of 0.9, 1.7, 3.4, 6.9, 13.8 and 27.5 μM were applied in the electrochemical system, respectively, and the peak current value gradually increased with the increase in the epinephrine concentrations (Figure 5C). The plot of the oxidative peak current (@ 0.40 V) against EP concentration takes two linear relationships as shown in Figure 5D.In each instance, the resultant linear regression equation has a favorable linearity of 0.9998. The limit of detection (LOD) was calculated by the detection limit formula, and the LOD value was determined as 0.262 μM.

Differential pulse volts (DPV) testing was also employed to estimate the electrochemical sensing property of 1% Pd/$BiVO_4$ for EP detection in the same EP concentration range. The potential ranges from −0.2 V to 0.9 V, with a scanning rate of 1.0 mVs$^{-1}$ and a pulse height of 50 mV. As a result, the oxidative peak current value gradually increases with the continuous increase in the EP concentration, and the $C_9H_{13}NO_5$ oxidation peak appears at 0.35 V (Figure 6A). The oxidative peak current shows two good linear relationships to the EP concentration from 0.9 to 3.4 μM and 6.9 to 27.5 μM accordingly with the linear regression equations of $I_{pa}$ (μA) = 0.2632 C (μM) + 0.597 ($R^2$ = 0.9998), $I_{pa}$ (μA) = 0.1153 C (μM) + 1.1007 ($R^2$ = 1.00), respectively (Figure 6B). DPV presents a better linearity than the CV method and the detection limit is calculated as 0.154 μM.

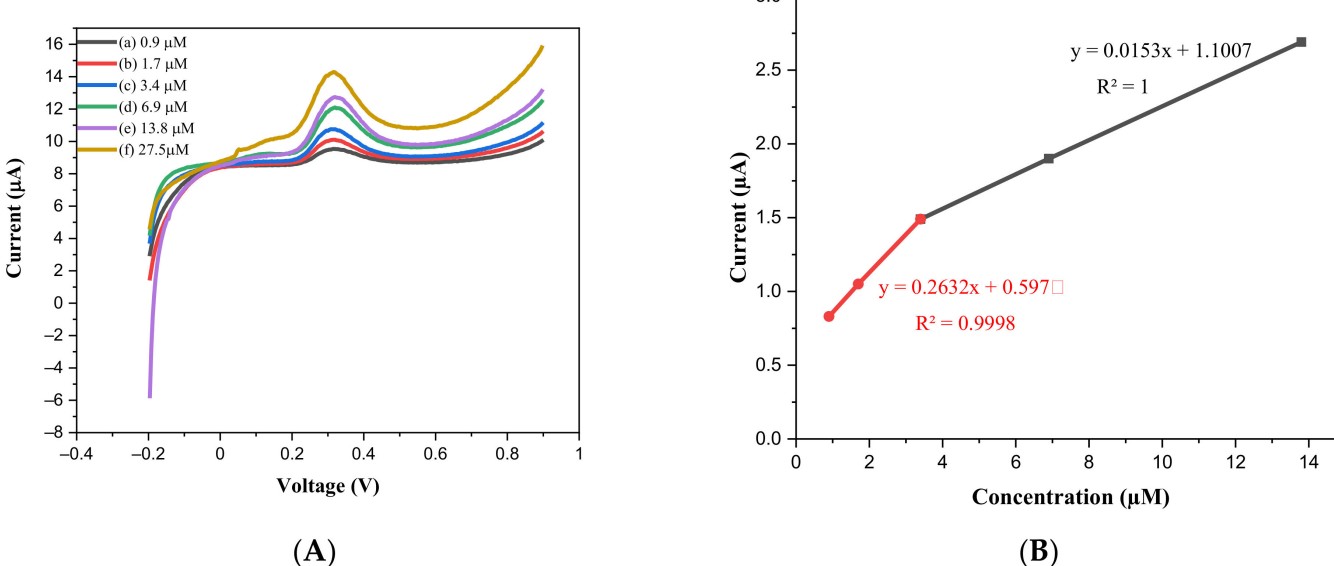

**Figure 6.** (**A**) DPV I-V curve of 1%Pd/BiVO₄/GCE with various concentrations of epinephrine (a) 0.9 μM (b) 1.7 μM (c) 3.4 μM (d) 6.9 μM (e) 13.8 μM (f) 27.5 μM; (**B**) The plot of DPV current against various epinephrine concentrations at 1%Pd/BiVO₄/GCE.

The repeatability of the 1% Pd/BiVO₄/GCE was investigated by continuously measuring DPV responses with the same electrode in the presence of 0.9 μM EP for 15 days in Figure 7. The obtained $I_{pa}$ values were maintained around 1.6 μA for about half a month, suggesting that the proposed electrodes have an excellent repeatability. A long-time stability of 1% Pd/BiVO₄ in the detection of EP has been proven in Figure 7.

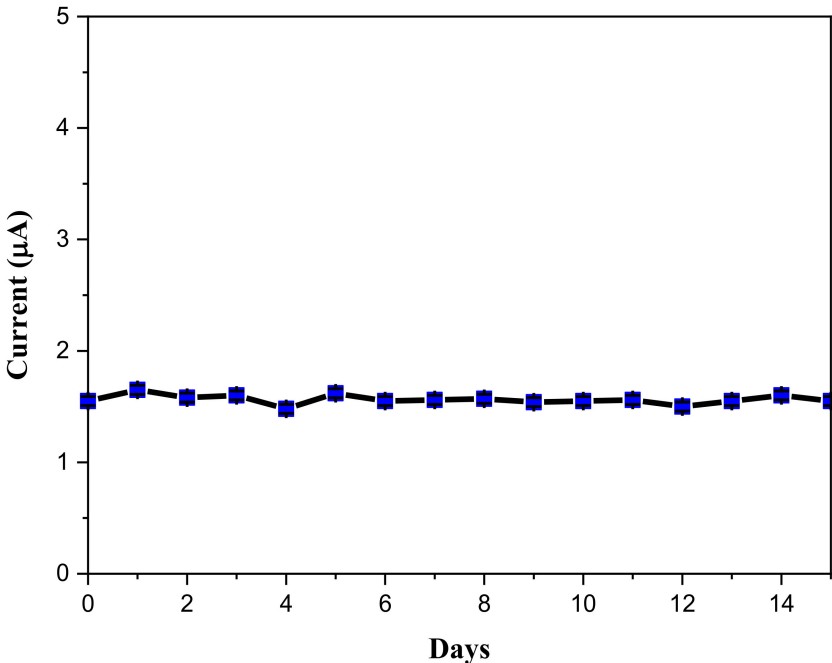

**Figure 7.** The stability test of the 1% Pd/BiVO₄/GCE on 3.4 μM EP.

The interference test of 1% Pd/BiVO₄ electrode was performed, and the results are shown in Figure 8a,b. The influence of biological and pharmaceutical interferences on the epinephrine CV oxidation signal was studied in the presence of 3.4 μM ascorbic acid, L-cysteine, and dopamine at 0.42, 0.38, and 0.22 V, respectively. Figure 8 reveals that the CV currents to 3.4 μM of ascorbic acid, L-cysteine, dopamine and epinephrine were obtained as

0.09, 0.14, 0.13 and 1.25 µA, respectively. The current signal of epinephrine was almost ten times greater than the interferences. These results confirm that the 1% Pd/BiVO$_4$ electrode has good selectivity for the determination of epinephrine.

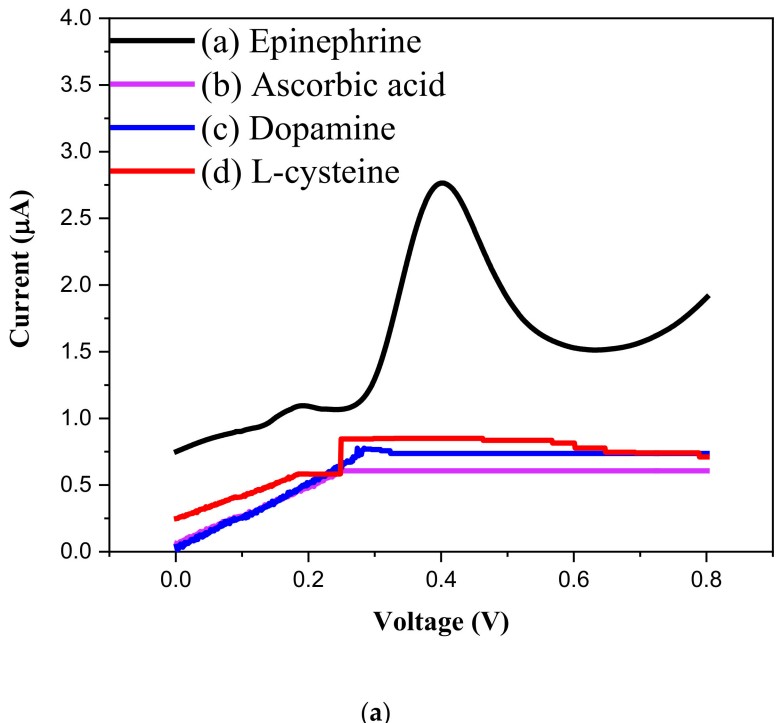

(**a**)

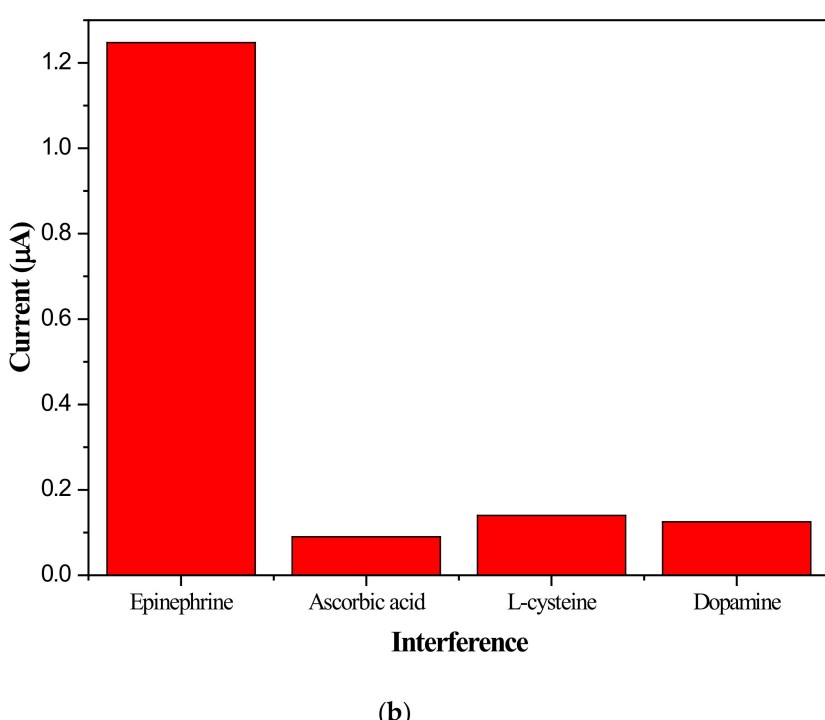

(**b**)

**Figure 8.** (**a**) and (**b**) Interference test of CV current to 3.4 µM ascorbic acid (0.42 V), L-cysteine (0.38 V) and dopamine (0.22 V) comparing to epinephrine (0.40 V).

### 2.3. Sensing Mechanism

The synthesized 1%Pd/BiVO$_4$ was used as sensing material for epinephrine detection in electrochemical system. The electrocatalytic redox mechanism of EP at 1%

Pd/BiVO$_4$/GCE is presented in Figure 9. The sensing materials catalyzed the oxidation of EP to form epinephrine quinone, as illustrated in Equation (1) [18–20].

$$C_9H_{13}NO_3 => C_9H_{11}NO_3 + 2\,H^+ + 2e^- \tag{1}$$

**Figure 9.** Electrochemical oxidation mechanism of epinephrine on Pd/BiVO$_4$/GCE modified electrode.

The improvement of the electrochemical sensing performance might be attributed to the highly dispersed Pd nanoparticles on the surface of BiVO$_4$ efficiently collecting the EP oxidation-induced electrons during the electrocatalytic reaction to transfer to BiVO$_4$ and then the surface of GCE.

In the meantime, the presence of an appropriate amount of Pd (1%) nanoparticles can accelerate the electrocatalysis-induced electron–hole pair separation and superoxide radical formation [18]. Appropriate amounts of additional Pd can promote the sensing signals; the overloading palladium of 2% might block the surface-active sites of BiVO$_4$. Therefore, the 1% Pd/BiVO$_4$ composite exhibits a quicker electrochemical detection of epinephrine.

### 3. Experiment

#### 3.1. Materials

Bismuth(III) nitrate pentahydrate (Bi(NO$_3$)$_3$·5H$_2$O, 98%), palladium (II) nitrate (Pd(NO$_3$)$_2$, 99.9%), ammonium metavanadate (NH$_4$VO$_3$, >99%), and ethylene dinitrilote-traacetic acid (EDTA, 99%) were purchased from Sigma-Aldrich. Nitric acid (HNO$_3$, 78–80%), sodium hydroxide (NaOH, 99.9%), and methanol (CH$_3$OH, 99.9%) were procured from J.T. Baker (Phillipsburg, NJ, USA). Epinephrine (98%, Aladdin Industrial Corporation (Shanghai, China)) and lidocaine were offered by Hualien Tzu-Chi Hospital. The above materials were used without further purification. Distilled and deionized (DI) water was used in the experiments prepared by a Milli-Q water purification system (Millipore).

#### 3.2. Preparation of Sensing Materials

The amount of 4.37 g Bi(NO$_3$)$_3$·5H$_2$O, 2.63 g EDTA and 1 ml HNO$_3$ were dissolved in 50 mL DI water with magnetic stirring at 90 °C for 30 min to form solution A, and 1.05 g NH$_4$VO$_3$ was dissolved in 50 mL DI water under continuous stirring for 30 min at 60 °C to form solution B. Solutions A and B were mixed together with vigorous stirring at 50 °C for 1 h. Following this, the mixed solution's pH value was adjusted to 7.0 by adding 1 M NH$_4$OH solution. After that, the solution was poured into a Teflon-lined stainless-steel autoclave and then transferred into an oven then heated at 180 °C for 6 h. The obtained

precipitate was filtered and washed with DI water and ethanol several times, and then dried at 100 °C for 12 h and calcined at 450 °C for 4 h to obtain pure $BiVO_4$ powder. The amount of 0.5 g $BiVO_4$ and appropriate amounts of $Pd(NO_3)_2$ were added into 100 mL DI water and stirred in an ice bath for 2 h; subsequently, 0.1 M $NaBH_4$ was added whilst stirring, then the mixture was stirred in an ice bath for 4 h. Finally, the precipitate was washed with ethanol, DI water, and then dried at 80 °C in an oven, until the dark green $Pd/BiVO_4$ powder was obtained.

### 3.3. Preparation of Pd-BiVO$_4$/GCE

The glassy carbon electrode (GCE, 3 mm diameter.) was first polished carefully with 0.01 mm α-$Al_2O_3$ powder and then a fine emery paper. $Al_2O_3$ and other residues were then removed by rinsing the surface with distilled water and ethanol, and then the electrode was sonic cleaned in distilled water and ethanol for 5 min, respectively. An appropriate amount of Pd-$BiVO_4$ with 5% chitosan and 5 mL acetone were mixed ultrasonically for more than 30 min. Then, around 50–75 μL of this slurry was dropped onto the freshly polished GCE via a syringe. After being dried in an oven at 80 °C, making the acetone evaporate completely, the prepared electrode (Pd-$BiVO_4$/GCE) was used as the working electrode.

### 3.4. XRD, SEM and TEM/EDX

The crystal structure of the Pd/BiVO4 composite material was characterized through X-ray diffractometer (Shimadzu XRD-6000, Cu X-ray tube (Cu Kα 1 = 1.54060 Å)). The characterization of the particle planes were conducted through XRD at a voltage of 40.0 keV, with the 2θ scanning range from 10 to 80°, at a scanning rate of 4 °/min. The surface microstructure of the samples was studied by a field emission scanning electron microscope (FE-SEM) (JEOL JSM-7000F). The morphology of the sensing materials was investigated through a high-resolution transmission electron microscope (HR-TEM) (JEM-2010). The elements of the samples were analyzed by an energy dispersive spectrometer (EDS) (England EXFORD Inca X-Stream).

### 3.5. Electrochemical Detection System

The electrochemical detection system is shown in Figure 1b. The potentiostat/galvanostat (PG-stat) (ZIVE SP1 compact type Electrochemical Workstation) was applied for the electrochemical measurements by cyclic voltammetry (CV) and differential pulse voltammetry (DPV) analysis. In this electrochemical system, three electrodes were used; $BiVO_4$- and Pd/$BiVO_4$-coated glassy carbon electrode (GCE) as the working electrode, a saturated Ag/AgCl electrode as the reference electrode, and platinum wire as the counter electrode.

## 4. Conclusions

In this research, the Pd/$BiVO_4$ composite was fabricated for electrochemical sensing of epinephrine and was characterized using XRD, SEM, TEM and EDS. High-purity $BiVO_4$ doping with Pd nanoparticles was obtained. The 1% Pd/$BiVO_4$-modified GCE electrode presented favorable electrocatalytic behavior using both CV and DPV methods with a good redox sensing signal. The concentration of epinephrine was determined with a linear range of 0.9 μM to 27.5 μM; high correlation coefficient ($R^2$ = 0.9998 for CV, 1.00 for DPV) and low limit of detection (0.262 μM for CV, 0.154 μM for DPV) were obtained. The proposed modified electrode also demonstrates excellent reproducibility, stability, and selectivity. Combing the simple fabrication process and low cost, 1% Pd/$BiVO_4$ has great potential as a sensing element in practical applications of epinephrine detection.

**Author Contributions:** Conceptualization, H.-N.L. and R.-J.W.; methodology, R.-J.W.; software, Y.-S.L.; validation, B.-H.H.; formal analysis, B.-H.H.; investigation, B.-H.H.; resources, B.-H.H.; data curation, B.-H.H.; writing—original draft preparation, R.-J.W.; writing—review and editing, H.L. and

R.-J.W.; supervision, H.-N.L. and T.-Y.C.; project administration, R.-J.W.; funding acquisition, H.-N.L. All authors have read and agreed to the published version of the manuscript.

**Funding:** This research was funded by Ministry of Science and Technology (Grant No.: MOST 110-2113-M-126-001), Taiwan.

**Acknowledgments:** The authors would like to thank the Ministry of Science and Technology (Grant No.: MOST 110-2113-M-126-001), Taiwan, R.O.C., for the financial support that it has provided for this study.

**Conflicts of Interest:** The authors declare no conflict of interest.

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
