# Peer review of "Promotion Effect of Palladium on BiVO4 Sensing Material for Epinephrine Detection"

_catalysts, doi:10.3390/catal11091083_

Round 1
Reviewer 1 Report
Comments to the author:
In this review the authors reported the fabrication of a Pd/BiVO4 nanocomposite as an electrochemical sensor of epinephrine. The authors reported the whole characterization of the composite with different loadings of Pd and they performed the sensing by two electrochemical techniques, cyclic voltammetry, and differential pulse voltammetry. Overall, this contribution is well written, the composites are well characterized, and the chemical sensing is satisfactory. I would recommend the publication of this contribution and I added just a few minor issues should that be addressed.
The following are some questions and suggestions for improving their work:
Major issues:
- As compared to the other nanomaterials reported in Table 1 the performance of the composite of the authors does not present an exceptional performance (in terms of sensitivity). So, could the authors highlight which are the advantages of their sensor?
- It would be interesting, as the authors prepared materials with different amount of Pd that, the electrochemical properties were reported
- Could the authors provide the CV curves of the other interfering agents and not only the histogram of figure 8?
Minor issues:
- SEM image of 2% Pd/BiVO4 is missing.
Author Response
The following are some questions and suggestions for improving their work:
Major issues:
- As compared to the other nanomaterials reported in Table 1 the performance of the composite of the authors does not present an exceptional performance (in terms of sensitivity). So, could the authors highlight which are the advantages of their sensor?
Response: We are thankful for the suggestion of the reviewer. We have added novelty by following in Table 1 and in the revised manuscript.
The highlights of the sensor
- Simple prepared sensing material and easy fabrication on electrode.
- Excellent sensitivity and selectivity.
- Good reproducibility and stability.
“Therefore, the object of this work is to develop a simple prepared BiVO4 based electrochemical sensor for EP detection. The electrochemical detection system is displayed in Fig. 1(b). The fabricated Pd doped BiVO4 was prepared onto a carbon paste electrode as the working electrode.” (line 74-78)
“The proposed Pd/BiVO4 is a promising sensing material for EP detection with a wide range and low detection limit as presented in Table 1.” (line 79-81 in the revised manuscript)
- It would be interesting, as the authors prepared materials with different amount of Pd that, the electrochemical properties were reported, Could the authors provide the CV curves of the other interfering agents and not only the histogram of figure 8? Minor issues: SEM image of 2% Pd/BiVO4 is missing
Response: We are thankful for the suggestion of the reviewer. We have added CV curves of the other interfering agents in Fig 8A in the revised manuscript. Sorry for the missing of the SEM image of 2% Pd/BiVO4. Since the 1% Pd/BiVO4 in this study exhibited the best sensing performance, we only listed the results of 1% Pd/BiVO4 when discussing some of the material properties.

Reviewer 2 Report
The research article ‘Promotion Effect of Palladium on BiVO4 sensing material for Epinephrine Detection’
In this research, the Pd/BiVO4-coated glassy carbon electrode was used for electrochemical detection of Epinephrine by cyclic voltammetry and differential pulse voltammetry methods. The Authors aim is to show that a Pd doped BiVO4 structure formed on a glassy electrode could be favourable for the detection of Epinephrine in comparison to other structures used for this purpose. In the introduction part, it is overviewed the need for sensitive methods for the detection of epinephrine. Electrochemical sensing is highlighted as a cheaper and easier method to perform in comparison to other most common methods.
The manuscript is of high quality and should be accepted after a minor revision.
It should be made some improvements:
- There is a mistake in line 94: “pH value was adjusted to 1 by adding 1 M NH4OH solution“. What was the actual pH? Please, clarify.
- Fig 3 might be improved. EDX mapping should be added to show where Pd is located. Might it be, that BiVO4 has Palladium not only on top of the structure but as well in the inner structure? Might it be a cause of the better performance of 1% Pd sample?
- Please make all numbers in all figures bigger and easier to spot.
Author Response
- There is a mistake in line 94: “pH value was adjusted to 1 by adding 1 M NH4OH solution“. What was the actual pH? Please, clarify.
Response: We are thankful for the suggestion of the reviewer. We have change the wrong typing to correct.
“And then, the mixed solution’s pH value was adjusted to 7.0 by adding 1 M NH4OH solution.” (line 218)
- Fig 3 might be improved. EDX mapping should be added to show where Pd is located. Might it be, that BiVO4has Palladium not only on top of the structure but as well in the inner structure? Might it be a cause of the better performance of 1% Pd sample?
Response: We are thankful for the suggestion of the reviewer. We have change the figure 3 in the revised manuscript, and the EDX mapping result ( as shown below) has been added. And we also revised some texts according to reviewer’s comments.
- Please make all numbers in all figures bigger and easier to spot.
Response: We are thankful for the suggestion of the reviewer. We have change the figures in the revised manuscript.
- There is a mistake in line 94: “pH value was adjusted to 1 by adding 1 M NH4OH solution“. What was the actual pH? Please, clarify.
Response: We are thankful for the suggestion of the reviewer. We have change the wrong typing to correct.
“And then, the mixed solution’s pH value was adjusted to 7.0 by adding 1 M NH4OH solution.” (line 218)
- Fig 3 might be improved. EDX mapping should be added to show where Pd is located. Might it be, that BiVO4has Palladium not only on top of the structure but as well in the inner structure? Might it be a cause of the better performance of 1% Pd sample?
Response: We are thankful for the suggestion of the reviewer. We have change the figure 3 in the revised manuscript, and the EDX mapping result ( as shown below) has been added. And we also revised some texts according to reviewer’s comments.
- Please make all numbers in all figures bigger and easier to spot.
Response: We are thankful for the suggestion of the reviewer. We have change the figures in the revised manuscript.
- There is a mistake in line 94: “pH value was adjusted to 1 by adding 1 M NH4OH solution“. What was the actual pH? Please, clarify.
Response: We are thankful for the suggestion of the reviewer. We have change the wrong typing to correct.
“And then, the mixed solution’s pH value was adjusted to 7.0 by adding 1 M NH4OH solution.” (line 218)
- Fig 3 might be improved. EDX mapping should be added to show where Pd is located. Might it be, that BiVO4has Palladium not only on top of the structure but as well in the inner structure? Might it be a cause of the better performance of 1% Pd sample?
Response: We are thankful for the suggestion of the reviewer. We have change the figure 3 in the revised manuscript, and the EDX mapping result ( as shown below) has been added. And we also revised some texts according to reviewer’s comments.
- Please make all numbers in all figures bigger and easier to spot.
Response: We are thankful for the suggestion of the reviewer. We have change the figures in the revised manuscript.
- There is a mistake in line 94: “pH value was adjusted to 1 by adding 1 M NH4OH solution“. What was the actual pH? Please, clarify.
Response: We are thankful for the suggestion of the reviewer. We have change the wrong typing to correct.
“And then, the mixed solution’s pH value was adjusted to 7.0 by adding 1 M NH4OH solution.” (line 218)
- Fig 3 might be improved. EDX mapping should be added to show where Pd is located. Might it be, that BiVO4has Palladium not only on top of the structure but as well in the inner structure? Might it be a cause of the better performance of 1% Pd sample?
Response: We are thankful for the suggestion of the reviewer. We have change the figure 3 in the revised manuscript, and the EDX mapping result ( as shown below) has been added. And we also revised some texts according to reviewer’s comments.
- Please make all numbers in all figures bigger and easier to spot.
Response: We are thankful for the suggestion of the reviewer. We have change the figures in the revised manuscript.
